

# Predicting the potential distribution of *Phacellanthus tubiflorus* (Orobanchaceae): a modeling approach using MaxEnt and ArcGIS

Cheng Chang[1,2], Fengkun Cai[1,2], Lu Shen[2], Xiang Jia[2], Zhiguo Liu[1], Chenlu Wang[3], Yujie Fu[3] and Yumei Luo[2]

[1] The College of Chemistry, Chemical Engineering and Resource Utilization, Northeast Forest University, Harbin, China
[2] Jilin Provincial Joint Key Laboratory of Changbai Mountain Biocoenosis and Biodiversity, Changbai Mountain Academy of Sciences, Antu, China
[3] The College of Biological Sciences and Technology, Beijing Forestry University, Beijing, China

## ABSTRACT

*Phacellanthus tubiflorus* Sieb. et Zucc, a vascular plant species, is believed to possess pharmacological properties including anti-fatigue and immunoenhancement. However, its distribution data is limited. Owing to the prospective medicinal relevance of this species, we proposed a comprehensive investigation for conservation and utilization. In this study, we aimed to scrutinize the plant holistically, ranging from the macroscopic to microscopic level. Specifically, we developed an ecological model using 51 records of *P. tubiflorus* subjected to seven environmental conditions. This model attained an exceptional area under curve (AUC ) value of 0.990 with a standard deviation of 0.004, and true skill statistic (TSS) value of 0.989, indicating a potently predictive capacity. Through the MaxEnt model, we completed a systematic depiction of the ecological niche of *P. tubiflorus*, revealing its primary global distribution. We carried out field surveys in the Changbai Mountain region to validate the model's accuracy and conducted observations focusing on the phenological attributes of *P. tubiflorus*, highlighting its largely subterranean existence. Factors such as seasonality of precipitation and temperature were found to sway its distribution, engendering comparably stable acclimation habitats. This research contributes to the data repository for facilitating subsequent studies on this species. Integrating botanical and ecological approaches, we proposed a more profound comprehension and evaluation of a species' behavior, survival strategies, and associations with other populations within specific habitats. Furthermore, this inclusive approach would assist in addressing pivotal environmental issues related to species conservation, biodiversity, and land development.

Corresponding authors
Chenlu Wang, greatwall1ok@163.com
Yujie Fu, yujie_fu@163.com

## INTRODUCTION

*Phacellanthus tubiflorus* Sieb. et Zucc. is a parasitic herbaceous vascular plant with significant ecological and medical importance. It is primarily distributed in temperate regions of Northeast Asia, including China, Russia, and South Korea (*Jung et al., 2013*; *Xu et al., 1991*). The species relies on broad-leaved host trees, such as oak and maple, for water and nutrients, utilizing specialized haustoria to establish a parasitic connection (*Jung et al., 2013*; *Yoon, Shin & Yi, 2013*). This unique relationship significantly impacts forest ecosystems by influencing host plant populations, nutrient cycling, and interspecies competition (*Kuijt, 1969*; *Westwood et al., 2010*). In addition, species of the Orobanchaceae family, sequester secondary metabolites from their host plants (*Press & Phoenix, 2005*). These sequestered compounds are believed to contribute to the pharmacological activities of the species, as supported by numerous studies on parasitic plants (*Genovese et al., 2021*; *Heide-Jørgensen, 2008*). In northeastern China, *P. tubiflorus* has been traditionally valued for its tonic and invigorating properties, with emerging research highlighting its potential use in treating fatigue and enhancing vitality, thus underscoring its significance in natural medicine (*Xu et al., 1991*). However, the increasing demand for this plant in pharmacological applications and its role in forest ecosystems raise concerns about its conservation status and sustainable utilization.

The distribution and abundance of *P. tubiflorus* are likely influenced by both climatic and anthropogenic factors, similar to other parasitic plants (*Chung, Hsu & Peng, 2010*; *Li et al., 2020*; *Wang et al., 2023*; *Xiang, Lei & Zhang, 2016*). Studies on related parasitic species have shown that climatic variables, such as temperature and precipitation patterns, can significantly affect host plant availability and the physiological adaptability of parasitic plants, thereby shaping their distribution (*Jung et al., 2013*; *Yoon, Shin & Yi, 2013*). Moreover, habitat alterations due to deforestation, urbanization, and agricultural activities could reduce the availability of suitable hosts or alter microhabitats essential for the growth and survival of *P. tubiflorus*. While direct evidence linking these factors to *P. tubiflorus* is currently lacking, the species' dependency on specific host plants and environmental conditions suggests its potential vulnerability to such changes. This underscores the need for targeted studies to explore the interactions between *P. tubiflorus*, its host plants, and the broader environment under varying climatic and anthropogenic scenarios.

To address these challenges, ecological research increasingly employs species distribution models (SDMs) to predict potential distribution patterns under current and future climatic scenarios (*Li et al., 2020*; *Xiang, Lei & Zhang, 2016*). These models integrate environmental data and known species distributions to identify areas suitable for survival and growth, offering insights into habitat dynamics and range shifts (*Marsh et al., 2023*; *Zhang et al., 2022*). In this study, we applied the MaxEnt model to predict the global distribution of *P. tubiflorus* and conducted field investigations in Changbai Mountain, Northeast China. By analyzing its ecological niche and environmental adaptability, we aim to provide a comprehensive understanding of the factors influencing its distribution and a theoretical foundation for conserving *P. tubiflorus* populations.

## MATERIALS & METHODS

The study area for modeling primarily covers the full distribution range of *P. tubiflorus* in Northeast Asia. This area represents the ecological and phenological conditions across the species' broad geographic range.

### Geographic distribution data

The geographic distribution data of *P. tubiflorus* primarily originated from two sources: (1) the Chinese Virtual Herbarium, a domestic specimen database (https://www.cvh.ac.cn/) (*Lu et al., 2018*); (2) the Global Biodiversity Information Facility (GBIF, https://www.gbif.org/) (*Benavides Rios et al., 2024*; *Feng et al., 2024*; *Lessa et al., 2024*). The latitude and longitude information obtained from these two sources was used as input data for MaxEnt modeling to predict the potential distribution of *P. tubiflorus*. There were 51 occurrence records of *P. tubiflorus* and the occurrence records of its host tree species, including 792 records of *Pinus koraiensis*, 2,231 records of *Quercus mongolica*, 275 records of *Tilia mandshurica*, 368 records of *Fraxinus mandshurica*, 145 records of *Euonymus pauciflorus*, and 660 records of *Tilia amurensis*, all obtained from two secondary sources: (1) and (2). The inclusion of occurrence records for host tree species is essential to understanding the ecological relationship between *P. tubiflorus* and its host trees. This relationship plays a critical role in shaping the distribution of *P. tubiflorus*, and the data helps inform ecological modeling efforts by identifying key host species and their influence on the spread and survival of *P. tubiflorus* in different environments. These occurrences were documented in the Northeast Asian region (Fig. 1). Fundamentally, both *P. tubiflorus* and its hosts are predominantly distributed in Northeast Asia, encompassing China, the Korean Peninsula, and Japan.

### Environmental variable data

Environmental variable data includes climate and soil variables. The climate variables encompass 19 bioclimatic factors sourced from the WorldClim database (http://worldclim.org). Soil data is extracted from the "HWSDv1.2" dataset within the World Soil Database (https://www.fao.org/soils-portal/en/), encompassing 16 soil factors, such as annual mean temperature, topsoil gravel content (*Hao & Wu, 2023*; *Zhu et al., 2023*) (refer to Table 1). Utilizing ArcGIS 10.8 software, we processed all the environmental variables with uniform spatial coordinate systems (GCS-WGS84) and spatial resolution (30").

To mitigate the risk of multicollinearity among environmental factors leading to model overfitting and enhance prediction accuracy, a preliminary simulation experiment was conducted for *P. tubiflorus*. Simultaneously, distribution points were numerically sampled with environmental factors using ArcGIS (Table 1), and the obtained values undergo Pearson correlation analysis (*Campos et al., 2023*; *Hou et al., 2023*; *Li et al., 2024*).

By comparing the results of the preliminary simulation experiment and correlation analysis, variables with contribution rates below 1% in the preliminary MaxEnt model, which indicates their minimal contribution to the model's predictive power, and those with correlation coefficients $|r| > 0.8$ and low contribution rates, are excluded. Notably, bio_3 and bio_18 exhibit correlation coefficients $|r| > 0.8$, but they have higher contribution rates

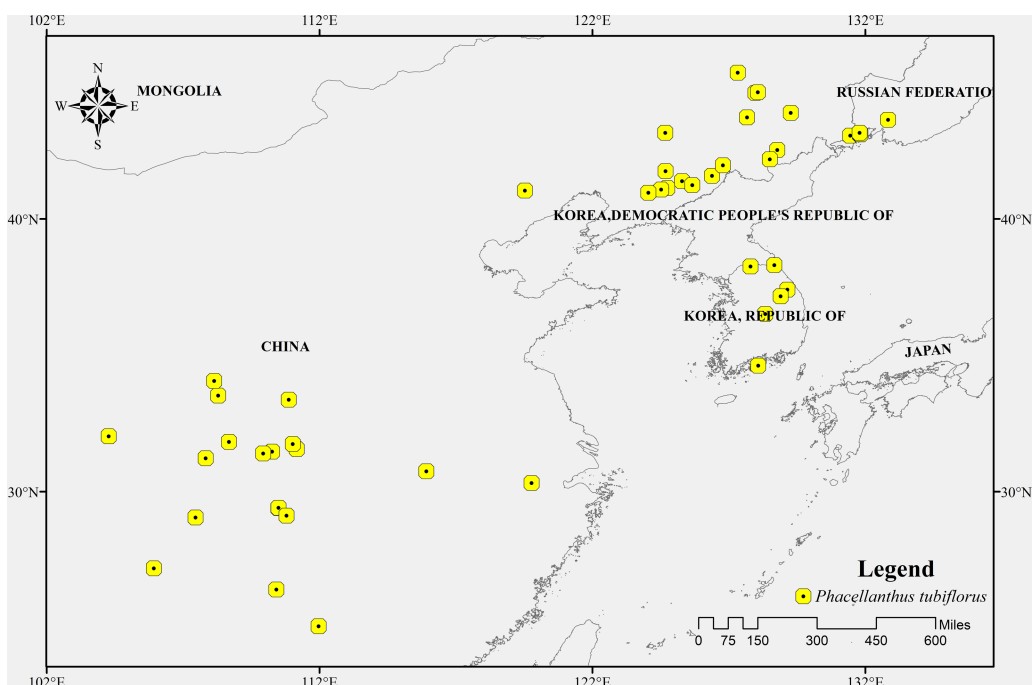

**Figure 1** **The occurrence records of *Phacellanthus tubiflorus*.** A major portion of the study area where presence data were collected.

in the experiment. Finally, seven variables, including precipitation of warmest quarter, Topsoil TEB, are chosen for predicting the suitable habitat of *P. tubiflorus* (Fig. 2).

## Model optimization and accuracy evaluation

The MaxEnt model estimates the maximum entropy distribution under environmental constraints, using precise species occurrence data and environmental variables. With minimal data requirements, the model only necessitates species occurrence coordinates and environmental variables, accommodating both continuous and categorical data and accounting for interactions between different variables. This versatility extends to constructing conditional models based on presence/absence data, mitigating data bias. The model generates continuous output results with a well-defined mathematical framework (*Phillips, Anderson & Schapire, 2006*). Consequently, the MaxEnt model is widely used to predict the potential distribution of various species, including plants (*Kunwar et al., 2023*), animals (*Berhanu, Tassie & Sintayehu, 2022*; *Liu et al., 2022*) and microorganisms (*Aidoo et al., 2022*; *Jie, Yang & Li, 2020*; *Yuan, Wei & Wang, 2015*).

The distribution data of *P. tubiflorus* and the environmental variables were input into the MaxEnt 3.4.1 model. Seventy-five percent of the distribution data was randomly allocated as the training set for habitat modeling, with the remaining 25% designated as the test set to evaluate model accuracy. We used the R package "sdmtune" to perform model optimization. We employed the jackknife method to analyze the impact of diverse environmental factors on distribution and generate corresponding response curves. The

**Table 1  Potential environmental factors affecting the distribution of *P. tubiflorus*.**

| Environmental variable | Interpretation | Environmental variable | Interpretation |
|---|---|---|---|
| bio_1 | Annual mean temperature (°C) | bio_19 | Precipitation of coldest quarter (mm) |
| bio_2 | Mean diurnal range (Mean of monthly (max temp–min temp)) (°C) | sio_1 | Topsoil gravel content (% vol) |
| bio_3 | Isothermality (bio_2/bio_7) ($\times$100) | sio_2 | Topsoil sand fraction (% wt) |
| bio_4 | Temperature seasonality (standard deviation $\times$100) | sio_3 | Topsoil silt fraction (% wt) |
| bio_5 | Max temperature of warmest month (°C) | sio_4 | Topsoil clay fraction (% wt) |
| bio_6 | Min temperature of coldest month (°C) | sio_5 | Topsoil USDA texture classification |
| bio_7 | Temperature annual range(bio_5- bio_6) (°C) | sio_6 | Topsoil reference bulk density (kg dm$^{-3}$) |
| bio_8 | Mean temperature of wettest quarter (°C) | sio_7 | Topsoil organic carbon (% weight) |
| bio_9 | Mean temperature of driest quarter (°C) | sio_8 | Topsoil pH (H$_2$O) (-log(H+)) |
| bio_10 | Mean temperature of warmest quarter (°C) | sio_9 | Topsoil CEC (clay) (cmol kg$^{-1}$) |
| bio_11 | Mean temperature of coldest quarter (°C) | sio_10 | Topsoil CEC (soil) (cmol kg$^{-1}$) |
| bio_12 | Annual precipitation (mm) | sio_11 | Topsoil base saturation (% ) |
| bio_13 | Precipitation of wettest month (mm) | sio_12 | Topsoil TEB (cmol kg$^{-1}$) |
| bio_14 | Precipitation of driest month (mm) | sio_13 | Topsoil calcium carbonate (% weight) |
| bio_15 | Precipitation seasonality (Coefficient of variation) (mm) | sio_14 | Topsoil gypsum (% weight) |
| bio_16 | Precipitation of wettest quarter (mm) | sio_15 | Topsoil sodicity (ESP) (%) |
| bio_17 | Precipitation of driest quarter (mm) | sio_16 | Topsoil salinity (Elco) (dS m$^{-1}$) |
| bio_18 | Precipitation of warmest quarter (mm) | | |

model underwent 10 iterations, and the average of the calculated values was deemed the final prediction result. Assessment of the model's accuracy utilized the receiver operating characteristic curve (ROC) and AUC value. A higher AUC value, approaching 1, signifies more accurate prediction results. The output ASC files from MaxEnt were imported into GIS, converted into grid data, and overlaid onto the global grid.

MaxEnt delineates the current suitable habitat areas of *P. tubiflorus* on a global scale. It employs the "Conversion tools-ASCII to Raster" functionality in ArcGIS 10.8 software to transform the prediction results from "asc" format files to raster files. These raster files undergo classification using the Natural Breaks (Jenks) method to stratify the present suitable habitat areas of *P. tubiflorus* into distinct levels. Specifically, based on MaxEnt prediction, the suitability of *P. tubiflorus* habitat is categorized into different levels. The GIS's reclassification function categorized regions into four suitability grades based on the suitability index (P) (*Mahatara et al., 2021*). The suitability levels are determined according to the probability of occurrence (P): $P < 0.3$ designates unsuitable habitat, $0.3 \leq P < 0.5$ indicates low-suitability habitat, $0.5 \leq P < 0.7$ represents moderate-suitability habitat, and $P \geq 0.7$ (*Cao et al., 2022*; *Renjana et al., 2022*) signifies high-suitability habitat.

Specifically, the model was evaluated using the area under the curve (AUC) and the true skill statistic (TSS), which are commonly used metrics to assess the predictive performance and reliability of ecological niche models. These two indices are widely recognized as the most respected dimensionless indicators for verifying the accuracy of species distribution

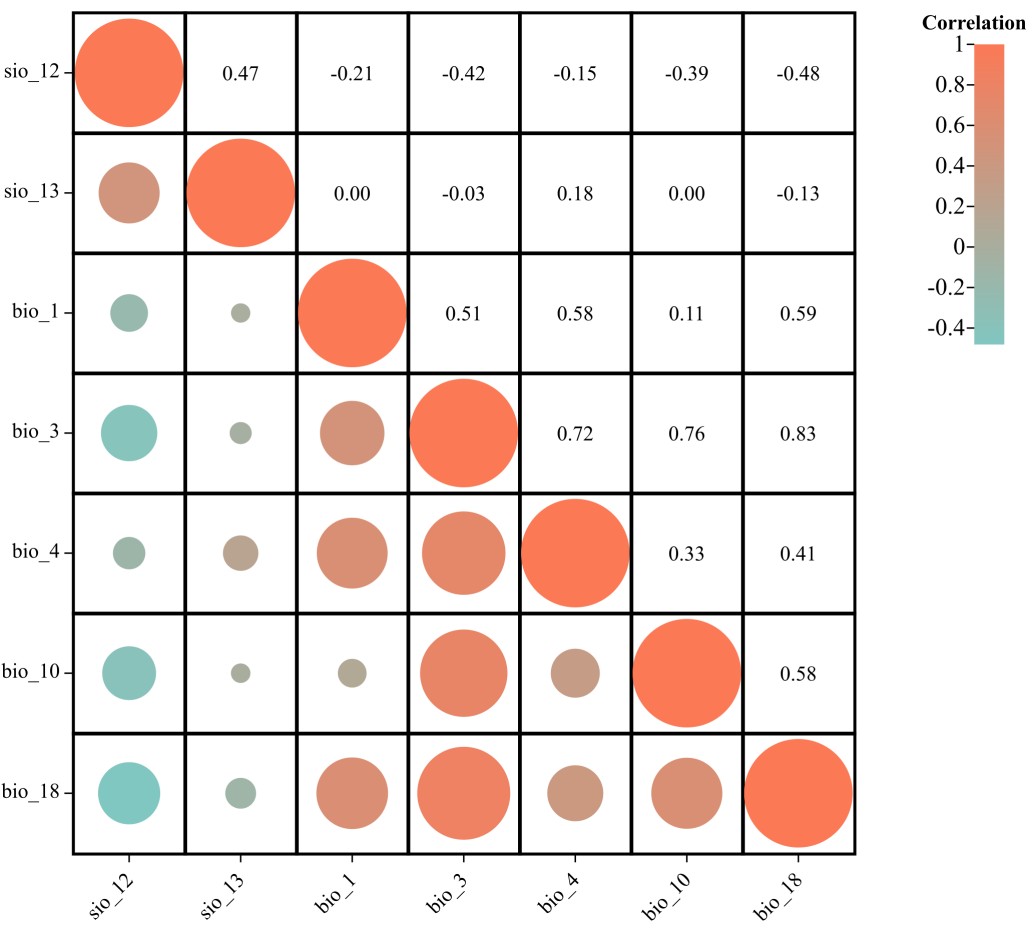

**Figure 2** Correlation analysis of environmental variables.

models (SDMs). TSS (*Allouche, Tsoar & Kadmon, 2006*) is a metric used to evaluate the predictive accuracy of ecological models, calculated as: TSS=Sensitivity+Specificity−1. The values of TSS span between −1 and 1, the evaluation criteria for TSS were excellent, 1.0–0.85; very good, 0.7–0.85; good, 0.55–0.7; fair, 0.4–0.55; and fail, <0.4; and $0.5 \leq AUC <0.6$, the simulation is considered unsuccessful; $0.6 \leq AUC <0.7$ indicates poor simulation performance; $0.7 \leq AUC <0.8$ suggests moderate simulation performance; $0.8 \leq AUC <0.9$ implies good simulation performance, and $0.9 \leq AUC <1$ indicates excellent simulation performance (*Chen et al., 2023*; *Renjana et al., 2022*; *Shi et al., 2023*; *Ye et al., 2022*).

The contribution ratio and permutation importance of environmental factors influencing the distribution of *P. tubiflorus* in its suitable habitat were analyzed using the 'knife-cutting method' within the MaxEnt model. The knife-cutting method, also known as the jackknife test, evaluates the relative importance of environmental factors by analyzing the change in model performance when each variable is either excluded or used in isolation. This approach helps determine the key factors driving species distribution (*Merow, Smith & Silander Jr, 2013*; *Phillips, Anderson & Schapire, 2006*).

### Field survey & validiation

After predicting suitable zones using the MaxEnt model, we conducted field sampling studies of *P. tubiflorus* to both validate our predictions and collect samples for subsequent biological research. The Changbai Mountain National Nature Reserve, located in northeastern China, was chosen for its diverse environmental conditions, which align with the habitat preferences of the target species. The field sampling aimed to gather specimens of *P. tubiflorus* for validating the model's prediction accuracy.

Following the sampling methods outlined by scholars (*Elith & Leathwick, 2009*; *Guisan & Thuiller, 2005*; *Kearney & Porter, 2009*), we divided the study area into three large regions based on the geographical area of the Changbai Mountain Conservation Area. Within each region, three groups of field workers conducted random sampling. As the objective was to collect *P. tubiflorus*, the sampling process was biased towards areas with higher suitability for the species. Each group collected data from three sampling points, after which sampling was concluded.

Additionally, geographical coordinates were recorded at each validation point. All field experiments were approved by the Changbai Mountain Protection & Development Management Committee.

## RESULTS

### Model optimization and accuracy assessment of the predicted potentially suitable area of *P. tubiflorus*

Following 10 repeated runs, the average AUC and TSS value for the current suitable habitat model of *P. tubiflorus* is determined to be 0.990 and 0.989, respectively, and the standard deviation is 0.004, signifying excellent predictive performance of the model (Fig. 3).

### The primary environmental factors influencing the distribution of *P. tubiflorus*

Refer to Table 1 for all the environmental variable information. When considering only one variable, the variables with a gain greater than 1 are precipitation of the warmest quarter and isothermality (Fig. 4).

Climate factors exert the most significant impact on the distribution of *P. tubiflorus*, constituting a total contribution ratio of 91.7% (Table 2). Precipitation of the warmest quarter and temperature seasonality hold substantial contribution values, reaching 60.3% and 23.9%, respectively. Mean temperature of the warmest quarter has a lower contribution ratio of only 1.6%. Annual precipitation and precipitation of the wettest month contribute 6.2% and 3%, respectively. In addition to climate factors, soil factors contribute 8.2% to the distribution of *P. tubiflorus*. Permutation importance reflects the model's dependency on environmental variables, where precipitation of the warmest quarter holds a substantial value of 63.9%, and isothermality follows with 26.1%. This indicates that the suitable habitat of *P. tubiflorus* is primarily influenced by precipitation of the warmest quarter, temperature seasonality, and annual precipitation, with precipitation and temperature being the key variables affecting its distribution, followed by soil properties.

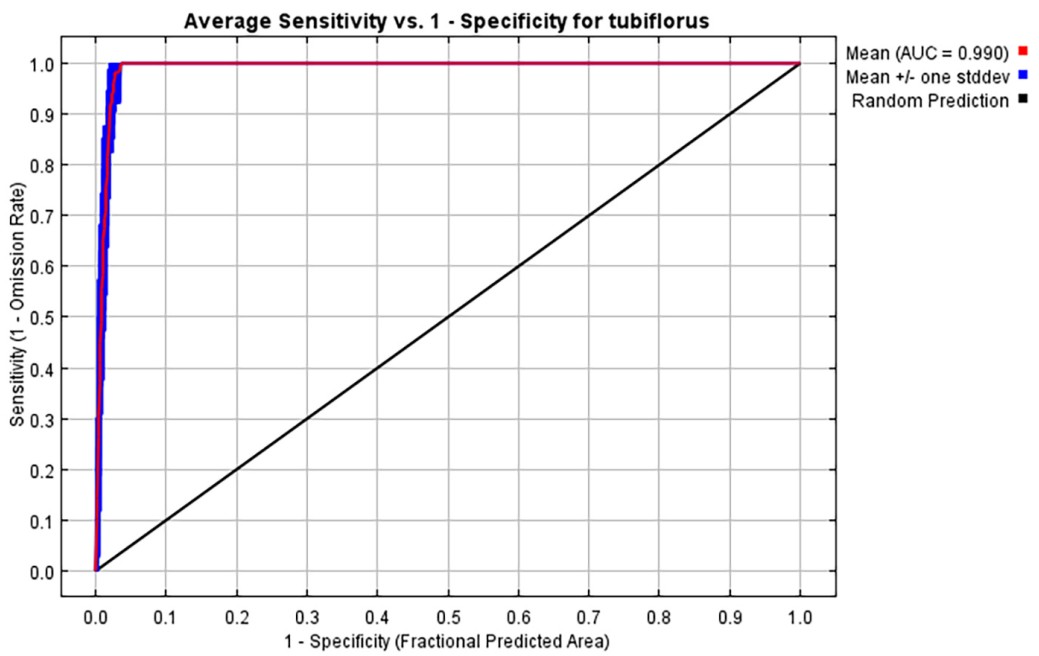

**Figure 3** ROC curves of the MaxEnt model for *P. tubiflorus* distribution.

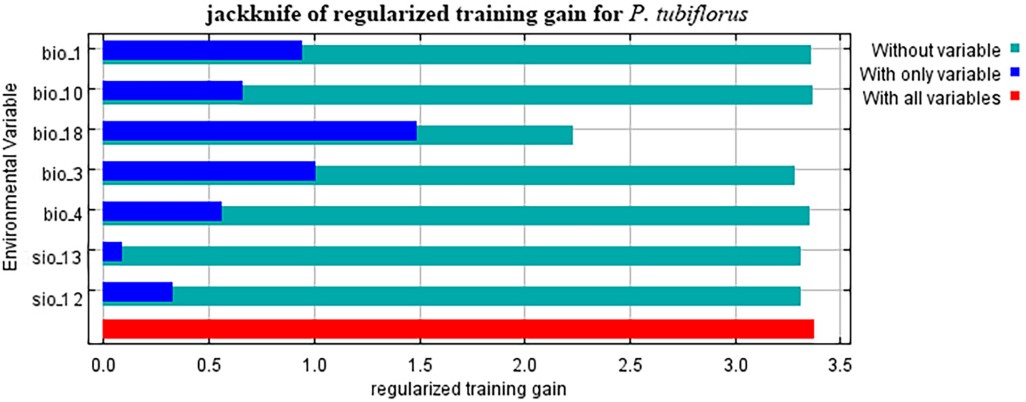

**Figure 4** The importance variables that affecting *P. tubiflorus*.

**Table 2** The contribution of environmental variables to the predicted distribution of *phacellanthus tubiflorus*.

| variable | bio_18 | bio_4 | sio_12 | bio_1 | bio_3 | sio_13 | bio_10 |
|---|---|---|---|---|---|---|---|
| Percent contribution (%) | 60.3 | 23.9 | 6.2 | 3.8 | 2.1 | 2 | 1.6 |
| Permutation importance (%) | 63.9 | 1 | 1.3 | 5.1 | 26.1 | 1.7 | 0.9 |

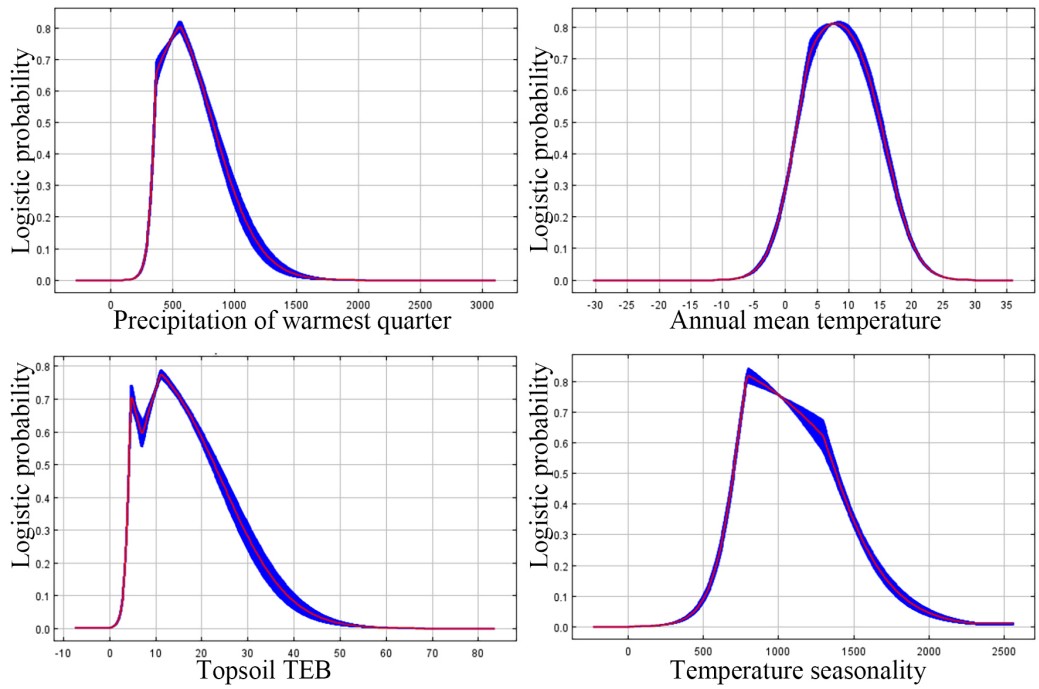

**Figure 5   The major response of variables that affecting *P. tubiflorus*.**

The probability of *P. tubiflorus* exhibits a unimodal pattern akin to a normal distribution (Fig. 5), correlating with the increase in warmest quarter, temperature seasonality, topsiol TEB, and annual mean temperature.

The threshold values of predominant environmental factor influencing the distribution of the suitable habitat for *P. tubiflorus*: precipitation of warmest quarter 330 mm to 860 mm, annual mean temperature 3 °C to 15 °C, topsoil TEB 3.4 (cmol kg$^{-1}$) to 22 (cmol kg$^{-1}$), and temperature seasonality 700 to 1,350 (Fig. 5).

## The suitable habitat distribution

The potential global distribution of *P. tubiflorus* within various countries worldwide was determined by the MaxEnt model, as illustrated in Fig. 6. The suitable habitat of *P. tubiflorus* encompasses an extensive area of 2,408,898 km$^2$, with high-suitability habitats comprising 804,034 km$^2$, constituting around 33.4% of the total suitable habitat area (Fig. 7). China contains the largest distribution of high-suitability areas, spanning 364,620 km$^2$ and accounting for 45.3% of the total high-suitability habitat. Japan follows with 261,778 km$^2$ (32.6%), and North Korea ranks third with 71,677 km$^2$ (8.9%). Despite their smaller land areas, Japan, North Korea, and South Korea exhibit high-suitability proportions relative to their territories, reaching 69.3%, 58.3%, and 49.9%, respectively (Fig. 8). Russia and Norway also feature significant areas of high-suitability habitat (Table 3). The moderate-suitability habitat covers 766,637 km$^2$, while the low-suitability habitat spans 838,314 km$^2$, accounting for 34.8% and 31.8% of the total area, respectively. Notably, the low-suitability habitat displays the broadest distribution.

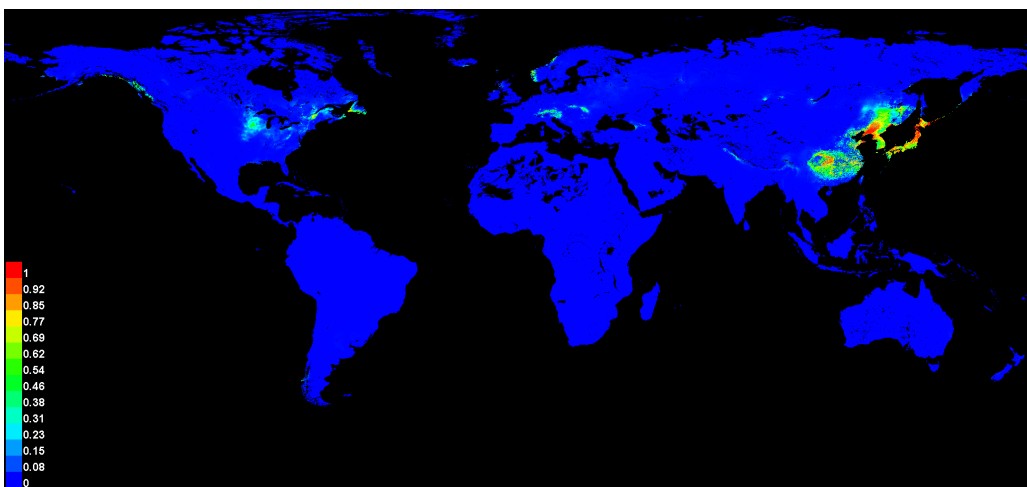

**Figure 6** **The global suitable habitats provided by MaxEnt models of *P. tubiflorus*.** The regions with the grid cells that indicated *P* value.

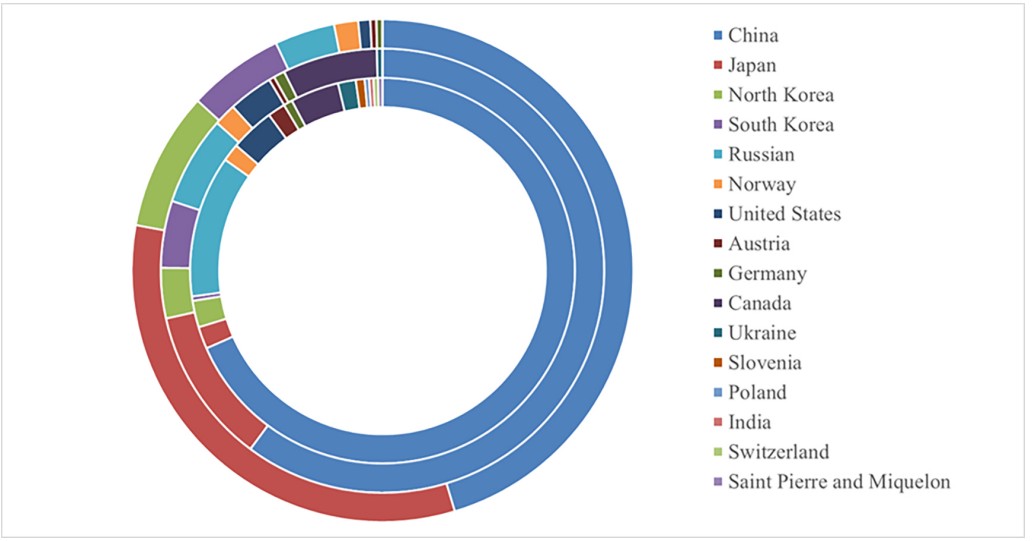

**Figure 7** **The global the potential distribution of *P. tubiflorus* around the world.** From inside out on the circle: low suitable habitats, moderate suitable habitats, high suitable habitats.

## Field assessment

Nine sampling points containing *P. tubiflorus* were surveyed, covering high, medium, and predicted suitable habitats based on the distribution model (Figs. 9 and 10). This strong association with specific host species highlights its intricate ecological relationships within forest habitats.
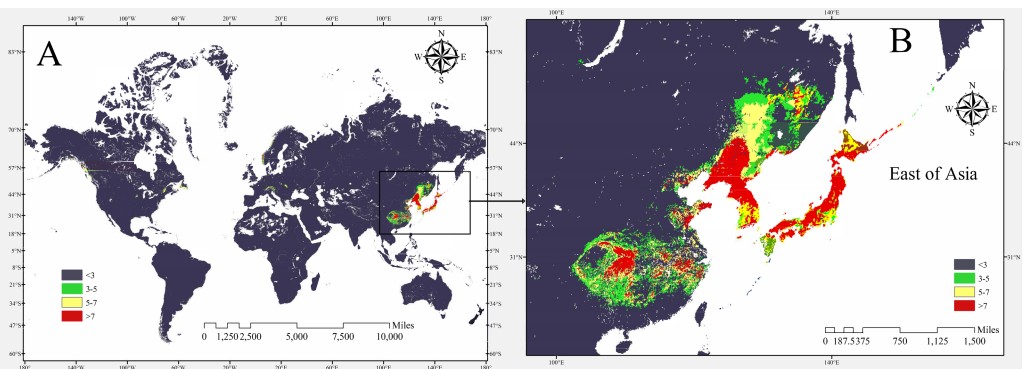

**Figure 8** The potential distribution of *P. tubiflorus* in the major region.

**Table 3** Statistics of the potential distribution Area of *P. tubiflorus* by countries.

| Country | Low-suitability habitats area (km²) | Middle-suitability habitats area (km²) | high-suitability habitats area (km²) |
|---|---|---|---|
| CHINA | 573,419 | 461,229 | 364,620 |
| JAPAN | 15,582 | 87,259 | 261,778 |
| South korea | 18,698 | 28,048 | 71,677 |
| North korea | 3,116 | 37,397 | 49,863 |
| RUSSIAN FEDERATION | 99,725 | 49,863 | 31,164 |
| NORWAY | 12,466 | 12,466 | 12,466 |
| UNITED STATES | 31,164 | 24,931 | 6,233 |
| AUSTRIA | 12,466 | 3,116 | 3,116 |
| GERMANY | 6,233 | 6,233 | 3,116 |
| CANADA | 34,281 | 52,979 | 0 |
| UKRAINE | 12,466 | 3,116 | 0 |
| SLOVENIA | 6,233 | 0 | 0 |
| POLAND | 3,116 | 0 | 0 |
| SWITZERLAND | 3,116 | 0 | 0 |
| SAINT PIERRE AND MIQUELON | 3,116 | 0 | 0 |
| INDIA | 3,116 | 0 | 0 |

# DISCUSSION

## Key environmental variables influencing habitat suitability of *P. tubiflorus*

### Precipitation of the warmest quarter as a key variable

The precipitation of the warmest quarter (Bio_18) was identified as the most critical variable influencing habitat suitability, contributing 60.3% to the model's performance. This result aligns with studies on other parasitic or mycorrhizal-dependent plants (*Chen et al., 2024*), which emphasize the importance of water availability during critical growth periods. The warmest quarter typically corresponds to the active growing season of *P.*

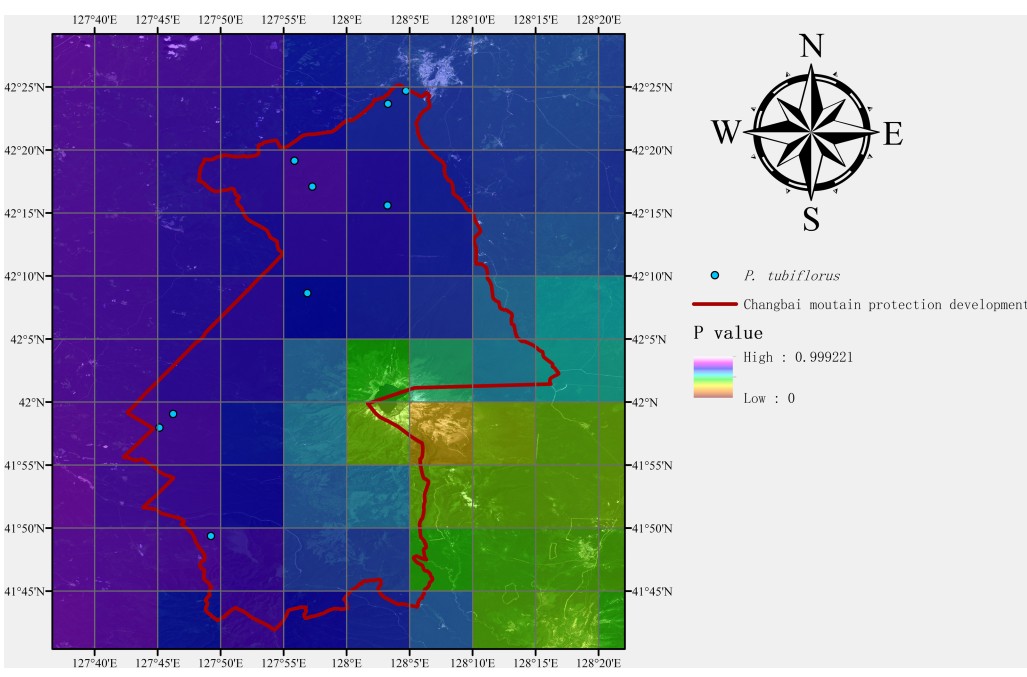

**Figure 9** **Field assessment of *P. tubiflorus*.**

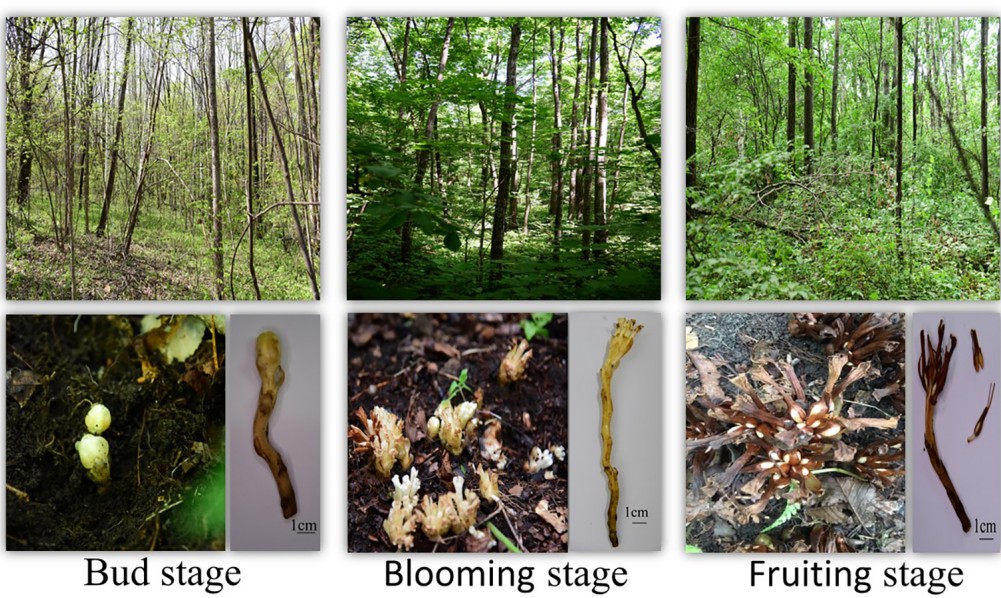

**Figure 10** **The phenological period of *P. tubiflorus*.**

*tubiflorus* and its host plants. Adequate precipitation during this period likely supports the underground development of its parasitic structures, including haustoria, which are essential for nutrient acquisition.

In Northeast Asia, where monsoon-driven precipitation patterns dominate, water availability during the warmest quarter could significantly influence the growth and survival of both *P. tubiflorus* and its host plants. In comparison, other studies on parasitic plants (*Těšitel, 2016*; *Watson, 2009*) have similarly highlighted precipitation's role as a limiting factor for distribution. These findings support the hypothesis that water availability during critical periods is a decisive factor for *P. tubiflorus*.

## Temperature seasonality (Bio_4)

Temperature seasonality (Bio_4) contributed 23.9% to the model's performance, making it the second most influential variable for the habitat suitability of *P. tubiflorus*. Temperature seasonality reflects the variation between the warmest and coldest months of the year, which plays a crucial role in defining the growing season and the timing of physiological processes such as flowering and fruiting. In regions with distinct seasonal temperature differences, *P. tubiflorus* may experience significant shifts in its life cycle, including changes in flowering time and nutrient acquisition, which are critical to its survival and reproduction.

Temperature seasonality has been identified as a key factor influencing the growth and distribution of other temperate-zone plants (*Hatfield & Prueger, 2015*; *Ida & Kudo, 2021*). This suggests that *P. tubiflorus* is sensitive to temperature fluctuations throughout the year, and such seasonal variation likely affects the species' ecological interactions and adaptation to different environments. Therefore, temperature seasonality reinforces the need to consider both temperature and precipitation as key climatic factors when modeling habitat suitability for *P. tubiflorus*.

## Topsoil TEB (Sio_12)

Topsoil TEB (Bio_12) contributed 6.2% to the model's performance, indicating a moderate effect of soil fertility on the distribution of *P. tubiflorus*. The availability of essential cations in the soil may influence the ability of the host plants to support parasitic interactions. *P. tubiflorus* depends on specific host plants for nutrient acquisition, and host plant growth is likely affected by soil nutrient levels, which in turn influences *P. tubiflorus*'s distribution.

Soil properties, including topsoil cation exchange capacity (TEB), have been shown to influence the growth and distribution of parasitic plants in other ecosystems (*Liang et al., 2009*; *Těšitel et al., 2021*). These findings suggest that *P. tubiflorus* prefers areas with soils that support the growth of host species capable of sustaining its parasitic lifestyle. The relatively moderate contribution of topsoil TEB to the model suggests that while soil fertility plays a role, it is not as critical as climate factors in shaping habitat suitability.

## Annual mean temperature (Bio_1)

Annual mean temperature (Bio_1) contributed 3.8% to the model's performance, making it the least influential among the key variables in determining habitat suitability for *P. tubiflorus*. While temperature can influence plant growth and physiological processes, its relatively low contribution suggests that *P. tubiflorus* may have a wider tolerance range

for temperature variability compared to precipitation. The impact of temperature could be more prominent in areas with extreme seasonal fluctuations, potentially influencing the species' growth patterns and reproductive cycles.

This result is consistent with studies on other temperate-zone parasitic plants, which have been found to be less sensitive to temperature fluctuations than to water availability during critical growth periods (*Hance et al., 2007*; *Watson, 2009*). The relatively minor role of annual mean temperature in this study suggests that *P. tubiflorus* may not be as temperature-sensitive as other species in the region, as long as it falls within an acceptable range for the host plants.

The model indicates that climate-related variables, particularly precipitation of the warmest quarter, temperature seasonality, and annual mean temperature, play significant roles in shaping the suitable habitat for *P. tubiflorus*. Precipitation of the warmest quarter is the dominant factor, but the contribution of other factors like temperature seasonality and topsoil TEB should not be overlooked. Together, these variables reflect the complex interplay of climate, soil, and host plant interactions that define the habitat suitability for *P. tubiflorus*. These results underscore the importance of considering multiple environmental factors when assessing the ecological needs of parasitic species and their distribution patterns.

## Understanding the suitability prediction for Northern Europe

The MaxEnt model predicted Northern Europe as a highly suitable area for *P. tubiflorus*, despite the absence of reported occurrences in this region. This discrepancy could arise from several factors, including ecological limitations not included in the model, such as soil composition, microclimatic variations, and symbiotic relationships with host trees. For instance, the species' strong dependence on host plants like *Quercus mongolica* and *Tilia mandshurica*, which are less prevalent in Northern Europe, may limit its establishment. Historical barriers, such as glaciation events and limited post-glacial recolonization, might also explain why the species has not dispersed to otherwise suitable areas (*Araújo & Guisan, 2006*; *Svenning, Normand & Kageyama, 2008*). This phenomenon is not unique to *P. tubiflorus*; similar findings have been reported for other plants, highlighting the importance of considering dispersal constraints and ecological niche conservatism (*Guisan & Thuiller, 2005*; *Thuiller et al., 2005*). These results emphasize that model predictions should be interpreted with caution when suitable areas remain unoccupied.

## Limitations and missing factors affecting the predicted distribution

In addition to ecological limitations, anthropogenic activities such as deforestation, urbanization, and agricultural expansion can significantly reduce the habitat of *P. tubiflorus* by altering forest structure and reducing host plant populations. Human-driven habitat fragmentation is a significant threat to parasitic plants due to the loss of suitable host species (*Carlson et al., 2020*). Incorporating land-use datasets in future models could offer a more comprehensive view of human impact on its distribution.

While *P. tubiflorus* is not strictly dependent on specific host species, as shown in this study, the diversity and abundance of host plants likely influence its habitat suitability. This

flexibility in host plant use aligns with patterns observed in other generalist parasitic plants (*Peterson et al., 2012*; *Watson, 2009*). However, the current model does not include host plant distribution data, and incorporating such data could improve the model's predictive power. Future models could benefit from integrating data on host plant diversity, collected through forest inventories or remote sensing.

Although elevation was considered in the model, finer-scale topographic features, such as slope and aspect, were not included. This exclusion was due to the spatial resolution of the available data and the representativeness of other environmental variables, such as temperature seasonality and precipitation, which already capture key climatic and soil conditions. While slope, aspect, and other topographic features, including sunlight exposure and soil moisture, are critical for the growth of parasitic plants (*Lu, Jiang & Zhang, 2022*), future studies utilizing higher-resolution datasets may incorporate these factors to further refine predictions of habitat suitability. The inclusion of high-resolution digital elevation models (DEMs) could significantly enhance prediction accuracy.

## CONCLUSIONS

This study employed the MaxEnt model to predict the potential distribution of *P. tubiflorus*, revealing its primary occurrence in Northeast Asia and offering valuable insights for conservation and resource utilization. Despite the limited sampling points, the model exhibited robust predictive performance (AUC: 0.990, TSS: 0.989), with key regions of suitability identified in China, the Korean peninsula, Japan, and smaller areas in Russia and Norway. The distribution is predominantly influenced by climatic factors, with precipitation of the warmest quarter contributing 60% to the model, underscoring the species' reliance on water availability during critical growth periods. These findings affirm the model's reliability for species distribution modeling and provide a foundation for guiding subsequent field surveys.

The study further highlights *P. tubiflorus*'s dependence on specific environmental conditions and its parasitic relationship with host plants, whose availability and distribution are critical to its survival. Consequently, conservation strategies should prioritize protecting suitable habitats while integrating host plant distributions into management plans to enhance protection efficacy. Looking ahead, future research should focus on expanding field data collection, particularly in under-sampled regions, and investigating additional ecological drivers—such as soil properties and interspecific interactions—that may refine habitat suitability assessments. Moreover, evaluating the potential impacts of climate change on *P. tubiflorus* and its hosts will be essential for long-term conservation planning and anticipating shifts in species distributions.

### Funding

This work was supported by National Key R&D Program of China (2023YFD2201801), the Special Forest Resources Discipline Innovation and Introduction Base at Northeast

Forestry University and 5.5 Engineering Research & Innovation Team Project of Beijing Forestry University (No: BLRC2023A01). The funders had no role in study design, data collection and analysis, decision to publish, or preparation of the manuscript.

### Grant Disclosures

The following grant information was disclosed by the authors:
National Key R&D Program of China: 2023YFD2201801.
The Special Forest Resources Discipline Innovation and Introduction Base at Northeast Forestry University and 5.5 Engineering Research & Innovation Team Project of Beijing Forestry University: BLRC2023A01.

### Competing Interests

The authors declare there are no competing interests.

### Author Contributions

- Cheng Chang conceived and designed the experiments, performed the experiments, analyzed the data, prepared figures and/or tables, authored or reviewed drafts of the article, and approved the final draft.
- Fengkun Cai analyzed the data, prepared figures and/or tables, and approved the final draft.
- Lu Shen performed the experiments, authored or reviewed drafts of the article, and approved the final draft.
- Xiang Jia performed the experiments, authored or reviewed drafts of the article, and approved the final draft.
- Zhiguo Liu conceived and designed the experiments, analyzed the data, authored or reviewed drafts of the article, and approved the final draft.
- Chenlu Wang conceived and designed the experiments, prepared figures and/or tables, and approved the final draft.
- Yujie Fu conceived and designed the experiments, analyzed the data, authored or reviewed drafts of the article, and approved the final draft.
- Yumei Luo analyzed the data, prepared figures and/or tables, and approved the final draft.

### Field Study Permissions

The following information was supplied relating to field study approvals (i.e., approving body and any reference numbers):
Field experiments were approved by the changbai mountain protection development management committee

### Data Availability

The raw measurements are available in the Supplementary Files.

## Supplemental Information

Supplemental information for this article can be found online at http://dx.doi.org/10.7717/peerj.19291#supplemental-information.

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
