# Peer review of "Predicting the potential distribution of *Phacellanthus tubiflorus* (Orobanchaceae): a modeling approach using MaxEnt and ArcGIS"

_PeerJ, doi:10.7717/peerj.19291_

## Round 0.1 · original submission · Major Revisions

Medicinal plants hold significant potential as sources for new medicines. Therefore, I believe your study could provide valuable insights for growers and plant collectors in identifying more suitable habitats for Phacellanthus tubiflorus. However, it is important to address certain technical aspects to enhance the clarity of your article. I strongly recommend carefully reviewing the reviewers' suggestions and thoughtfully considering each recommendation. If you disagree with any suggestion, it would be helpful to provide clear, well-reasoned justifications for your viewpoint. Additionally, your article would benefit from linguistic refinement. I suggest seeking assistance from a colleague or utilizing our editing service to ensure the language is polished and professional.

Reviewer 1 ·

Basic reporting

The idea seems naïve but the way of presenting and writing needs a lot of improvement. The authors are just throwing the sentences randomly throughout the text which lacks cohesion. More importantly, the introduction and discussion sections do not hold clear information, and the readers only guess what the study is about after reading the methods and results.
The writing needs to be reviewed by a professional English reviewer.

Experimental design

Methods seem good but the research questions are not well-defined and some of the results presented in the result section are not described clearly in the methods.
Overall, the concept is ok but the presentation is poor.

Validity of the findings

The findings are trustworthy but need to be redone by incorporating some suggestions made in the method section.
The validation approach is new, but it is a bit directionless which should be addressed in the revised version.

Additional comments

There is no clear input to the next step and a lot of information (for example, the interpretation of results) is lacking.

Annotated reviews are not available for download in order to protect the identity of reviewers who chose to remain anonymous.

Reviewer 2 ·

Basic reporting

The English needs improvement from a native speaker.
Introduction, Discussion and Conclusions should be better organized and more informative.

Phacellanthus is a hemiparasitic plant which sequesters secondary metabolites from their host plants. Thus host plants determine the metabolic profile and pharmacology of Phacellanthus. This topic needs to be explored in more detail.

Which secondary metabolites have been recorded from Phacellanthus? What are their pharmacological properties?

Experimental design

For such an analysis it is important to include all sites where the plant has been found.
Look here
https://www.worldfloraonline.org/taxon/wfo-0001089291
You all need to check all the herbaria, not only the Chinese Herbarium online

Validity of the findings

The quality of the modelling strongly depends on the input data. What is the coverage of the existing literature and herbarium data?

Additional comments

Title. tubiflorum not with capital
Distribution: check https://www.worldfloraonline.org/taxon/wfo-0001089291

---

## Round 0.2 · Major Revisions

I appreciate your constructive attitude toward the reviewers' suggestions and improving your article based on their suggestions. Although your article has been revised according to their suggestions, it needs more improvements. Carefully consider each recommendation, assessing its relevance and potential to enhance your work. If you find yourself disagreeing with any specific suggestion, it is important to provide clear and well-reasoned justifications, supported by evidence where applicable, to substantiate your perspective.

Reviewer 1 ·

Basic reporting

The study lacks a lot of information, the title and the body do not align with each other.

Experimental design

Not enough clarity.

Validity of the findings

Modelling work is valid. But I suspect the biological/phenological results as the authors do not provide the sufficient evidence for the drawn results.

Additional comments

The manuscript should be reviewed from professional language editor to minimize the redundancy.

Annotated reviews are not available for download in order to protect the identity of reviewers who chose to remain anonymous.

Reviewer 2 ·

Basic reporting

The revision has improved the first version.
In the introduction, you correctly mention that Phacellanthus is a parasitic plant with interesting medicinal properties. You need to mention that species of the Orobanchaceae take up secondary metabolites from their host plants (and not only nutrients)- there are many publications on this topic (check the literature).
These sequestered secondary metabolites are responsible for the pharmacological activities.

Experimental design

ok

Validity of the findings

ok

Additional comments

no

---

## Round 0.3 · Minor Revisions

I appreciate your positive and constructive attitude toward the suggestions of reviewers. However, your article needs some minor revisions to improve before publishing. Please carefully read the reviewer's comments and consider each of them. If you find yourself in disagreement with any particular suggestion, it would be beneficial to provide clear and well-reasoned justifications for your perspective.

Reviewer 1 ·

Basic reporting

First, I would like to thanks authors for incorporating all the suggestions and comments. The manuscript now looks well structured and scientifically valid. So, I suggest acceptance with one minor suggestion as below.
I am still not convinced with the title of the manuscript. To me, title is something that reflects the major focus of the study and more importantly, more than half of the readers decide either to read the manuscript or not solely on the basis of title. Thus, I do not think the word "phenological traits" used in the title is justifiable as there is no single description, results and discussion on this trait. So, my suggestion is to change the title that reflects the modeling work only.

Experimental design

no comment

Validity of the findings

no comment

Additional comments

no comment

Reviewer 2 ·

Basic reporting

The revision has improved the ms.
Several statements in the revised ms or not elegantly connected.

Experimental design

ok

Validity of the findings

ok

Additional comments

ok

---

## Round 0.4 · accepted · Accept

I would like to thank you for accepting the referees' suggestions and improving your article based on their suggestions. Your article is ready to publish. We look forward to your next article.

Reviewer 1 ·

Basic reporting

The title now reflects the main idea of the manuscript, thank you for the revision.

Experimental design

Ok

Validity of the findings

Ok

Additional comments

Ok